# Searching for Optimal Substitute Habitats for Plants by Biological Experiments—A Case Study of the Endangered Species *Aldrovanda vesiculosa* L. (Droseraceae)

**DOI:** 10.3390/ijerph191710743

**Published:** 2022-08-29

**Authors:** Magdalena Pogorzelec, Marzena Parzymies, Barbara Pawlik-Skowrońska, Michał Arciszewski, Jacek Mielniczuk

**Affiliations:** 1Department of Hydrobiology and Protection of Ecosystems, University of Life Sciences in Lublin, 20-262 Lublin, Poland; 2Institute of Horticultural Production, University of Life Sciences in Lublin, 20-612 Lublin, Poland; 3Department of Applied Mathematics and Computer Science, University of Life Sciences in Lublin, 20-612 Lublin, Poland

**Keywords:** environmental conditions, plant conservation, reintroduction, translocation, waterwheel plant

## Abstract

The selection of appropriate locations for the reintroduction of endangered plant species is an important process, because it usually influences the success of the conservation. The aim of this study was to select the optimal substitute habitats for *Aldrovanda vesiculosa*, taking into account the influence of physical–chemical factors (light intensity, temperature, pH, concentration of dissolved forms of nitrogen and cyanobacterial toxin microcystin-LR) on the efficiency of plant growth. Water analysis and field observations of the habitats of six lakes in Eastern Poland typified as potential substitute habitats for aldrovanda were carried out. The results of the experiments showed that both the concentration and the form in which nitrogen compounds are present in the environment were the factors limiting the growth rate and condition of plants. The second factor that caused the inhibition of aldrovanda growth was microcystin-LR. It was found that the habitat conditions in Lake Brzeziczno were within the ecological tolerance of the species. Particularly important was the low content of mineral compounds and the available forms of nitrogen and phosphorus in the water. Therefore, the probability of development of toxic cyanobacteria, the metabolites of which may affect the growth of *A. vesiculosa*, is also minimal.

## 1. Introduction

Among the many plant species threatened by extinction, endemic or relict species are the largest group. Currently, there are different attempts undertaken in many places around the world to protect them, by combining in situ and ex situ methods. There are also more and more reports on the translocations of plants, both for the reintroduction and strengthening of their populations [1,2,3,4,5,6,7].

Conservation translocation is the controlled placement of plants into natural or semi-natural habitats with the aim of preventing the extinction of species [8]. The experiences of both successful and unsuccessful reintroduction attempts conducted so far show clearly that knowledge of the species biology and the correct choice of habitat based on scientific data and practitioners’ assessments are extremely important while undertaking such activities [9,10,11,12].

A huge challenge is the translocation of aquatic species that can move in a habitat, because the physical–chemical conditions change dynamically, both in diurnal and seasonal rhythms. They are also very susceptible to the direct and indirect impacts of anthropogenic pressure. In such cases, the difficulty lies not only in the selection of the optimal replacement habitat for the species, but also in its subsequent monitoring and in the evaluation of the effectiveness of the activities carried out [13,14,15].

One of the species of which protection is necessary is *Aldrovanda vesiculosa* L. (waterwheel or aldrovanda). It is a very rare and critically endangered aquatic carnivorous plant [16]. Aldrovanda belongs to the Droseraceae family. It represents a monotypic taxon and is the only member of the genus. It was first established by Carolus Linnaeus in 1753, but it was first mentioned as *Lenticula palustris Indica* in 1696 in Plukenet’s collection [17,18]. In the past, the species was widely distributed in 43 countries in Africa, Asia, Australia, and Europe. Unfortunately, due to the rapid decline in the population, it has been reported as extinct from many regions of the world and assessed in the IUCN Red List with the category Endangered. The species is facing a potential threat in its natural habitat due to changing environmental factors such as eutrophication and climate alterations [18,19]. According to Cross [18], at the moment, aldrovanda is present in only about 50 stands worldwide. Most of them are located in Europe, despite the extensive degradation of its habitats and the disadvantageous changes in wetlands during the last years, such as afforestation, eutrophication, drainage, and intensive agriculture. Cross [18] stated 38 existing natural sites, with 90 unverified and 164 extinct. About 80 sites of the studied species were known in Poland in the past [18,19,20]. Recently, Kamiński [21] proved only nine natural sites with a continual occurrence, while Adamec [22] reported 14 natural and 4 artificial sites. *Aldrovanda vesiculosa* is a perennial, rootless, carnivorous plant. It floats just below the water surface in shallow, standing, and dystrophic waters [18,22]. It produces a poorly branched shoot, usually up to 20 cm long, with physiological polarity and steep structural growth [18,23,24]. Leaves grow in whorls and terminate into snapping traps, which catch prey with a rapid spectacular movement. Each trap consists of a two-lobed lamina with a midrib and 3–6 long bristles [18,25] (Figure 1). 

In favorable conditions, aldrovanda may flower and produce seeds, but the species typically propagates by apical branching of shoots [25,26,27]. In favorable conditions, lateral shoots are formed every 5–7 whorls, and adult plants usually have 2–4 branches. However, the shoots may be non-branched as well [22]. According to Adamec and Kovářová [28], branching frequency is around 5 days, and a high branching frequency indicates optimal ecological growth conditions. All lateral branches separate from the mother shoot, forming new plants or turions, which are vegetative dormant winter buds, formed as a response to lower autumn temperatures [13,29,30]. Turions are the shoot tips formed by the condensation of modified leaves, with short internodes. They are about 5–8 mm long and dark green [18,31,32]. They protect shoots tips against freezing [22]. 

*Aldrovanda vesiculosa* prefers to grow in shallow, standing water, with mesotrophic or dystrophic conditions and warm temperatures, high CO_2_ concentrations, and zooplankton as a food source. It is strongly sensitive to eutrophic conditions, and therefore it might be considered as a potential bioindicator [19]. 

Active conservation of the species has been conducted worldwide, and many scientists and practitioners have taken efforts to undertake different activities regarding rare and endangered carnivorous plants. As a part of conservations efforts, plants are introduced to new stands. Such actions were described by Lamont et al. [33], who introduced the Japanese race of aldrovanda to North America, or by Adamec [14] and Kamiński [20], who established new stands in the Czech Republic and Poland.

In order to preserve the genetic resources of *A. vesiculosa* in eastern Poland, about 10,000 plants were produced by the in vitro method with the purpose of translocated them into replacement sites or to strengthen the existing populations (as a part of the project “The active protection of *Aldrovanda vesiculosa* in the territory of Lubelszczyzna region”, no. POIS.02.04.00-00-0034/18). Therefore, in our work, we focused on selecting the most appropriate *A. vesiculosa* reintroduction site in eastern Poland in terms of habitat conditions. This is the first attempt to make such a selection on the basis of a series of biological experiments in which we tested environmental factors (e.g., light intensity, temperature, pH, dissolved nitrogen concentration) that represent the biggest changes and may be crucial for the survival of the plants after replacement. 

## 2. Materials and Methods

In the first stage, local investigations of the natural occurrence sites of *A. vesiculosa* in eastern Poland was carried out at Lakes Długie (D), Łukie (L), and Moszne (M), and the following reservoirs were selected as potential future replacement habitats for the species: Lake Brzeziczno (B), Lake Łukietek (LU), and small pond called Jelino (J) (Figure 2). In order to conduct laboratory experiments, in May 2019, lake water samples were taken from all of the above-mentioned locations. 

The following physical and chemical parameters of water were determined: total nitrogen, N_tot_ (photometric detection of generated nitrate after UV and thermos digestion via reduction to nitrite and azo dye formation according to DIN EN ISO 29441); the content of nitrites, N-NO_3_, and nitrates, N-NO_2_ (photometric via azo dye formation, according to DIN EN ISO 13395); ammonium nitrogen, N-NH_4_ (photometric by gas diffusion and color indicator, according to ISO 11732); total phosphorus concentration, P_tot_, and phosphates, P-PO_4_ (spectrophotometric method with ammonium molybdate); and dissolved organic carbon, DOC (with the use of a PASTEL UV analyzer). Electrolytic conductivity (µS·cm^−1^) and pH of the water samples were also determined using a YSI 556 MPS multi-probe field meter.

In March 2019, plant material in the form of *A. vesiculosa* turions was collected from two natural sites of the species located in the littoral zone of Lake Łukie and Lake Długie in Poleski National Park (Figure 3). Turions were deposited in filtered lake water, at a temperature below 10 °C, in the dark to inhibit their growth.

In order to determine the parameters of the physical–chemical factors of the environment that are optimal for the growth of plants from turions, experiments under laboratory conditions were carried out. The first test was performed to show the influence of different temperatures on that process (E1). An attempt was also made to determine the optimal light intensity for the effective growth of *A. vesiculosa* from turions (E2). Taking into account the different physical and chemical conditions of the reservoir waters, which can potentially be a place for the reintroduction of specimens, the significance of the water pH for the growth of *A. vesiculosa* plants from turions was also tested (E3). Due to the risk of a significant increase in the trophic status of the reservoirs into which *A. vesiculosa* may potentially be reintroduced, the influence of high concentrations of various forms of available nitrogen on the growth of plants from turions was estimated (E4). Due to the presence of the cyanobacterium *Microcystis aeruginosa* in the water of Lake Długie (data unpublished), an experiment was conducted to establish the influence of the cyanobacterial toxin microcystin-LR on turions and developed plants of *A. vesiculosa* (E5). To minimize failure in the reintroduction of the studied species, the last test was carried out to recognize the importance of the complex characteristics of the water for the efficiency of plant growth from turions (E6).

Immediately prior to each of the experiments, individual turions were gently blotted dry and weighed to determine their initial fresh weight (with an analytical scale). The turions were then placed individually in separate 5 mL chambers (20-chamber PCV plates). After completion of the experiments, the final fresh weight of each plant (mg F.W.), the length of its main and side shoots (mm), the number of leaf whorls, and the presence or absence of trap leaves were determined. The content of photosynthetic pigments, namely, chlorophyll *a*, *b* and carotenoids in fresh plants, before and after the completion of the selected experiments, was also determined. The weighed turions and plants were ground in an agate mortar in 1 mL of 96% ethyl alcohol. The ground material was transferred to Eppendorf tubes, which were then placed in a water bath for 5 min at a temperature below the boiling point of the alcohol (70–73 °C). Then, the test tubes were placed in an MPW-350 R centrifuge. The extract was centrifuged for 10 min (14,000 rpm at a temperature of 17 °C). The collected supernatant was cooled, and the absorbance measurements were done on a Specord 40 spectrophotometer at the three wavelengths of 665, 649, and 470 nm. The content of chl*a* and chl*b*, and total carotenoids in the fresh weight of plants were calculated according to the modified formulas proposed by Lichtenthaler and Wellburn [34]. 

### 2.1. Experiment 1 (E1)—The Effect of Temperature on Growth of Plants from Turions

The 36 *A. vesiculosa* turions from the L (Łukie) site were placed in the lake water obtained from Lakes Długie (D) (18 turions) and Lake Łukie (L) (18 turions). Plates with turions were placed in a growth cabinet under the conditions of continuous illumination with photosynthetically active light (70 µEm·m^−2^·s^−1^) for the period of 8 days in different thermal conditions of 10 °C, 22 °C, and 28 °C. Six turions were used each treatment.

### 2.2. Experiment 2 (E2)—The Effect of Light Intensity on the Growth of Plants from Turions

Fifteen *A. vesiculosa* turions from the L (Łukie) site were placed in the water of Lake Łukie (L) under the conditions of constant temperature (22 °C) and different light intensity: 25 µE·m^−2^·s^−1^; 50 µE·m^−2^·s^−1^; 70 µE·m^−2^·s^−1^, in a 12 h light/12 h dark photoperiod for 7 days. For each treatment, 5 turions were used.

### 2.3. Experiment 3 (E3)—The Effect of Water pH on the Growth of Plants from Turions

Twenty *A. vesiculosa* turions obtained from the L (Łukie) site were placed in the lake water of Lake Łukie (L) with different pH values, in the range of 4.0–8.0, and different electrolytic conductivities, and fixed with 0.01 M HCl or 0.01 M NaOH: pH 4.0 (443 µS·cm^−1^); pH 5.0 (399 µS·cm^−1^); pH 6.0 (352 µS·cm^−1^); pH 7.0 (307 µS·cm^−1^); and pH 8.0 (355 µS·cm^−1^). They were kept in a growth cabinet under the conditions of continuous illumination with photosynthetically active light (70 µEm^−3^·s^−1^) and constant temperature (22 °C). The experiment lasted 10 days, and in each treatment, 4 turions were used. 

### 2.4. Experiment 4 (E4)—The Influence of the Concentration of Two Dissolved Forms of Nitrogen on the Growth of Plants from Turions

Fifty-four *A. vesiculosa* turions from site L (Łukie) were placed in previously prepared solutions with various concentrations of N-NO_3_ and N-NH_4_. Solutions with different concentrations of nitrate nitrogen (N-NO_3_) and ammonium nitrogen (N-NH_4_) were prepared on the basis of filtered lake water (from Lakes Długie and Łukie), which was enriched separately with two forms of nitrogen, so that the total N concentration in each solution was 100 mg N·dm^−3^. KNO_3_ (potassium nitrate) and CH3COONH_4_ (ammonium acetate) solutions were used for that purpose. Then, two dilutions were made with distilled water from each of the prepared solutions (100 mg NO_3_ ·dm^−3^ and 100 mg NH_4_ ·dm^−3^ in water from Lake Długie and Lake Łukie), and the following solutions were obtained: 10 mg N·dm^−3^ and 1 mg N·dm^−3^. The water pH was adjusted to 7.0. Filtered lake water from both lakes—Długie and Łukie—was used as controls. The plates with turions were placed in a growth cabinet under illumination with photosynthetically active light (70 µE·m^−2^·s^−1^) in a 16 h light/8h dark photoperiod and constant temperature (22 °C) for 10 days. Average daily biomass and plant length increases were estimated, and the cormus biomass density was determined by calculating the ratio of the biomass to shoot length (g F.W./mm) of individual specimens.

### 2.5. Experiment 5 (E5)—The Influence of Microcystin-LR on Growth and Condition of Plants

Twelve turions (6 each from Lake Długie and Lake Łukie) and 24 *A. vesiculosa* plants developed from turions (12 each from Lake Długie and Lake Łukie) were used. They were placed in filtered lake water and in lake water from site L supplemented with two concentrations of pure microcystin-LR (purchased from ENZO): 100 and 200 μg·dm^−3^. Three turions and four plants were used per treatment, and they were placed in a growth cabinet in the conditions of constant temperature (22 °C) and light (70 µE·m^−2^·s^−1^), in a 12 h light/12 h dark photoperiod for 7 days. The concentrations of MC-LR higher than environmentally relevant were chosen for the experiments to observe more severe effects of cyanobacterial toxin while comparing different lake water.

### 2.6. Experiment 6 (E6)—The Influence of a Complex Physical–Chemical Factors of Various Aquatic Habitats on the Growth of Plants from Turions

*A. vesiculosa* turions were placed in the lake water of six reservoirs in which the studied species naturally occur (D, L, M), or which could potentially be reintroduction sites for the species (B, LU, J). Turions submerged in natural filtered water of the aquatic habitats (4 replicates in each) were placed in a growth cabinet under the conditions of constant temperature (22 °C) and light (70 µE·m^−2^·s^−1^) in a 12 h light/12 h dark photoperiod for 7 days.

### 2.7. Statistical Analysis

In the first stage of the analysis, the basic descriptive statistics were determined for the variables studied in the experiments at different levels of the experimental factors. Selected statistics were presented in comparative charts that show the mean value or median, and the range of min–max/range of variation. In the case of the E1–E3 experiments, the influence of the examined factor on the variability of the analyzed feature was examined. For that purpose, the non-parametric Kruskal–Wallis test was used. The analysis of variance with the HSD Tukey post hoc test was used to investigate the effect of the content of two available nitrogen forms on the growth of plants from turions (E4). When the distribution was not consistent with the normal distribution, the Mann–Whitney U test was used. In the next stage, the basic descriptive statistics for the parameters or physical–chemical factors of water at individual sites were calculated. The distribution of the tested parameters was presented with box-plots.

Two multidimensional methods were used for further analysis: hierarchical clustering (with the use of the average linkage agglomeration method and binomial index as a dissimilarity measure), and principal component analysis (PCA). In the multivariate analysis, both physical and chemical factors as well as plant growth efficiency, determined by biomass growth, and the length of shoots of individual specimens deposited in water from different reservoirs were taken into account. The results of the analyses were presented by means of a dendrogram and a PCA diagram (PCA biplot).

## 3. Results and Discussion

The results of research on the biology and ecology of *A. vesiculosa* carried out so far in various regions of the world have expanded the knowledge about this unique species. Much is already known about the methods of reproduction and dispersion, as well as the mechanisms and effectiveness of reducing available nitrogen deficiencies in the environment [27,35,36,37], although the question of the relationship between the size of aldrovanda plants and the amount and type of zooplankton captured has still not been resolved. Despite many reports on aldrovanda’s habitat requirements, the current problem is the correct selection of substitute habitats for the species. Due to the small and constantly decreasing number of sites of species occurrence in Europe and the limited number of populations, it is recommended to attempt its reintroduction or translocation to substitute habitats [20,22].

Actions aimed at active protection of the species must be based on strong scientific foundations, which increase their effectiveness and largely excludes possible failures. Whenever attempts are made to translocate plants, it is also necessary to recognize the specific habitat conditions for selected locations. An extremely important stage is the selection of habitats, which must refer not only to the physical–chemical and biocenotic factors shaping the habitat conditions, but also to changes that occur in the environment, as they may directly or indirectly affect the success of species reintroduction [38,39,40,41].

Therefore, to select a suitable habitat for the introduction of aldrovanda in eastern Poland, a series of experiments was carried out. The study used a limited number of plant materials in order to not over-exploit the population of that endangered species. In Table 1, the physical and chemical characteristics of lake water from where turions were collected are presented. 

At the beginning, the basic experimental conditions, i.e., temperature (E1) and light (E2), were established, which allow for optimal plant growth. The results of the first experiment showed that the distribution (median) of the tested feature, i.e., the total increase in plant biomass, during its growth from the turion differed significantly only in the case of the tested plants deposited in the water from Lake Długie (D), which grew at temperatures of 10 and 22 °C. In other cases, no statistically significant differences were found. Comparing the increase in biomass of plants from Lakes Długie and Łukie, statistically significant differences were found in the tested features at 10 and 28 °C. Plants from Lake Łukie grew more efficiently (Figure 4). No such relationship was observed when analyzing the results of the biomass increase of plants growing at 22 °C.

*Aldrovanda vesiculosa* inhabits reservoirs with low water levels, which heat up quickly in summer and do not freeze to the bottom in winter [22]. Adamec [19] and Adamec and Kovářová [28], while trying to determine the ranges of the values of physical and chemical factors of the environment for the proper functioning of the species, indicated the optimal temperature in the range of 23–26 °C. Ngangbam et al. [42] defined the most adequate temperature range for the growth of the tested species as 27–29 °C. However, it should be remembered that the period of growth of turion plants in the natural conditions of Eastern Europe takes place at a much lower temperature. The water temperature plays an important role in plant development, as it determines flowering, but when the water temperature is too high, it makes the plants fragile. Too low temperatures during the vegetation season may stop plant growth [27,43]. Therefore, it was decided that further experiments would be carried out at a constant temperature of 22 °C.

Light intensity is an important factor affecting the development of *A. vesiculosa*. It has been noticed that plants growing in sunny places tend to be smaller in length; however, they are characterized by a greater increase in biomass if compared with plants growing in shaded positions [20]. In the studies on the photosynthesis process in aldrovanda, Adamec [44] confirmed that it is a photophilic plant. The conducted analyses did not show a statistically significant effect of light intensity on any of the plant features studied in the second experiment (E2) (Figure 5). However, a tendency in the increased median values of plant growth features (biomass and number of leaf) with the increased light intensity was visible. There were also visible differences in the average content of photosynthetic pigments in the tested plants grown under different light conditions (Table 2). As the light intensity increased, the content of chlorophyll *a* and carotenoids increased as well. Therefore, in the next experiments, the constant light conditions of 70 µE·m^−2^·s^−2^ were maintained. There is no available information in the scientific literature so far on attempts made to establish the illumination requirements in a similar way. Other authors focused on Secchi disc visibility in water bodies, the color of the water, or the depth at which aldrovanda was found [19,28,42].

Aldrovanda’s ecological tolerance range is quite wide in terms of the physical and chemical parameters of the habitat. Studies conducted in Asia on the physiology of this plant show that it grows in slightly alkaline waters with pH values of 7.1 to 7.9 [17]. In European conditions, it most often inhabits slightly acidic reservoirs (pH 6.0–6.9) [20]. Most of the authors who examined the ecological tolerance of the species pointed to the optimal pH of the water for aldrovanda in the range of pH 5.7–7.6, although reports can also be found with a much wider range of pH 5.0–9.0 [13,25,26,27,44,45]. Adamec [29] indicates, for example, that aldrovanda developed properly in a habitat dominated by sphagnum mounds, where the pH ranged from 4.6 to 4.8.

The values of the water pH did not have a statistically significant influence on any of the plant features studied in our experiment. The values of the length increase (mm) of the tested plants were at the borderline of significance. It could be seen, however, that the highest values of biomass increase, length increase, and the number of whorls could be observed in plants kept in aqueous solutions with pH values not lower than 6.0 (Figure 6). 

The content of photosynthetic pigments in plants also turned out to be a feature that did not show a clear tendency to change at different pH values, although the average sum of pigments was much higher in those growing in water with a pH higher than 6.0 (Table 2).

According to Kamiński [46] the biggest threats to the Polish population of *A. vesiculosa* are the changes in the water level and the eutrophication of habitats. The obtained results in this study indicated that there was a visible influence of different nitrogen concentrations in the water on the morpho-physiological characteristics of *A. vesiculosa* (E4), especially during the tested stage—growing out of the turion. The analysis of the results showed that the form of the available nitrogen present in the water was more important than the amount of nitrogen in the solution itself. The process of plant development from turions was much more effective in nitrate nitrogen (N-NO_3_) solutions than in ammonium nitrogen (N-NH_4_) solutions. In solutions with N-NO_3_, a higher increase in plant length was observed, and a greater number of total trap leaves formed. As the concentration of N-NO_3_ in the solution increased, the plants grew more and more effectively. The results of the experiment also indicated that a low concentration of N-NH_4_ could stimulate the growth of plants from turions, while too high of a concentration was a factor limiting plant development (Figure 7 and Figure 8).

The cormus biomass density had higher values in water solutions with nitrate nitrogen, and the value of this parameter increased with increasing nitrogen concentration. In the case of plants kept in solutions containing ammonium nitrogen, the situation was the opposite—a decrease in “firmness” was noted with an increase in nitrogen concentration (Figure 7 and Figure 8). Apart from the influence on the physiological processes of plants, the increased content of biogenic elements in the water in an absorbable form also indirectly influences the populations of aldrovanda. The results of the “Monitoring of species and natural habitats of 2018” (Chief Inspectorate of Environmental Protection) indicate that the main threat to the species is the overgrowing of reservoirs by other submerged macrophytes and rush vegetation, which leads to a reduction in the area available for aldrovanda, and in the case of small reservoirs, to its complete disappearance.

Together with the eutrophication of surface freshwaters, mass development of bloom-forming cyanobacteria occur [47]. One of the most frequent and toxic cyanobacteria metabolites is MC-LR [48]. Therefore, the predictive influence of MC-LR on *Aldrovanda vesiculosa* was tested under experimental conditions in different lake waters. The results of experiment 5 (E5)—testing the influence of microcystin-LR on the growth and condition of plants—showed the influence of the toxic substance both on the growth of plants from turions and on the growth of fully developed plants exposed to its activity. There was an inhibition of growth and a decrease in biomass in relation to the control in all tested cases. It is worth noting, however, that the specimens of aldrovanda from Lake Długie, despite the effects of the toxic substance in various concentrations, grew more effectively than specimens from Lake Łukie, and the content of photosynthetic pigments in the fresh weight of those plants also increased (Figure 9; Table 1). The likely reason for this phenomenon is the fact that in the past in Lake Długie, the presence of *Microcystis aeruginosa* was recorded, the secondary metabolite of which is MC-LR (unpublished data). The *A. vesiculosa* population functioning in this reservoir had been previously exposed to the action of the toxin, which suggests the possibility of its adaptation and lowering of the level of physiological response to the presence of this chemical compound in the water.

The results of the last experiment to solve the most important issue concerning the selection of the optimal replacement habitat for *A. vesiculosa* in eastern Poland are presented in Figure 10 The results of the influence of the complex of natural physical and chemical factors of various aquatic habitats on the growth of turions into plants allowed for the preliminary indication of the most favorable location for introducing plants obtained from ex situ cultivation. Analyzing the results of the previously conducted studies on the values of selected physical and chemical factors in the water of six reservoirs, the differences between the lakes with natural aldrovanda populations and those selected for its reintroduction can be clearly seen (Figure 10).

An ordinance analysis (Figure 11) was also carried out, taking into account the complex of physical and chemical factors along with the plant growth efficiency, determined by the increase in biomass and the length of shoots of individual specimens kept in water from various reservoirs. The first two PCA axes explained 75.87% of the total variance in the dataset (PCA1: 45.03%; PCA2: 30.84%). EC (conductivity) and pH (acidity) seem to be the major contributors to the positive part of first component (PCA1), and TN (total nitrogen) to the negative part (Figure 11). The second principle component PCA2 was strongly positively associated with TP (total phosphorus). The biplot indicates a strongly negative correlation between TN and between TBI (total biomass increase) and SLI (stem length increase). Strong positive correlations were also observed between TBI and SLI; TP and DOC (dissolved organic carbon); and pH and EC. We found near-zero correlations between DOC with TBI and SLI.

In combination with the dendrogram analysis (Figure 12), it was found that the habitat conditions in the Brzeziczno (B) reservoir were within the ecological tolerance of the species and were statistically the closest to those considered as optimal (based on a comparison of the values of habitat conditions selected parameters of reservoirs where naturally populations of *A. vesiculosa* are functioning). Particularly important is the low concentration of mineral compounds in the water of Lake Rescigno (reflected in the low value of electrolytic conductivity) and low available forms of nitrogen and phosphorus (which proves a low trophic status). The probability of developing toxic cyanobacteria, the metabolites of which may negatively affect the growth of aldrovanda, is also very low. Microorganisms such as *Microcystis aeruginosa* rarely develop dense populations in oligotrophic and dystrophic reservoirs [49].

In the process of selecting substitute sites, the opinion of the practitioner also plays a very important role, taking into account phenomena observed in the field, not always measurable, but often allowing one to complete the characteristics of the habitat [11]. The chosen reservoir for reintroduction, Lake Brzeziczno, seems to be the most proper site for aldrovanda. Local inspection of the sites revealed that the littoral zone of Lake Brzeziczno is rich in small, shallow coves sheltered from wind and undulations and is inhabited by rush vegetation with a large share of sedges. Such conditions correspond to those described by Kamiński [46] as optimal for aldrovanda. According to this author, it forms associations with other plants found on the edges of lakes and peat bogs, being part of communities belonging to two phytosociological classes—*Potametea* and *Phragmitetea*. The largest clusters of the plant were recorded on the border of sedge phytocenoses from the *Magnocaricion* and *Nympheion* association and communities dominated by *Phragmites australis*, *Typha latifolia*, *Stratiotes aloides*, *Hydrocharis morsus-ranae* or *Nymphaea* sp., and *Nuphar lutea.*

## 4. Conclusions

Experiments on plants in laboratory conditions allowed the values of selected environmental factors to be indicated that are preferred by specimens of *Aldrovanda vesiculosa* originating from natural sites. The effective growth of the tested plants was observed under conditions such as temperature (oscillating in the range of 22–28 °C) and water pH (above pH 6). The limiting factors for the growth rate and condition of plants turned out to be not only the concentration of nitrogen compounds in the environment, but also, and perhaps even above all, the form in which they are found there. The negative effect on plants was especially observed in the presence of ammonium ions in the water. The second factor that inhibited the growth of aldrovanda was the cyanobacterial toxin microcystin-LR.

In the case of *A. vesiculosa*, it was noted that despite the large diversity of the habitat conditions in the studied sites, the coastal zone of Lake Brzeziczno may be considered as a proper replacement site for the population. First of all, the low water fertility of the reservoir, expressed by a low concentration of available nitrogen forms and low electrolytic conductivity, is not conducive to the excessive development of cyanobacterial populations, which are the source of toxins (i.e., microcystin-LR)—one of the most important factors that can inhibit the growth and development of aldrovanda in natural environments.

To sum up, it might be stated that both physical and chemical parameters of water together with requirements of the species, are crucial for the choice of the appropriate sites for plant translocation. However, the observations made by practitioners are also important, as they are able to recognize the problems not measurable by typical experimental methods.

## Figures and Tables

**Figure 1 ijerph-19-10743-f001:**
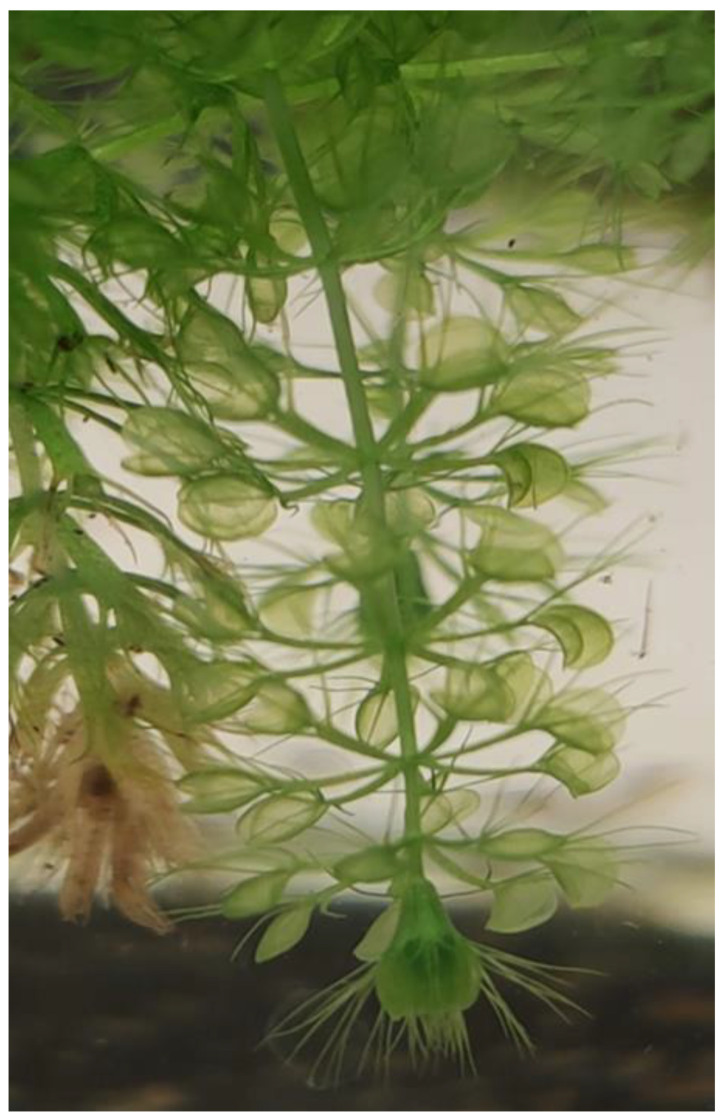
*Aldrovanda vesiculosa* L. shoot and leaves.

**Figure 2 ijerph-19-10743-f002:**
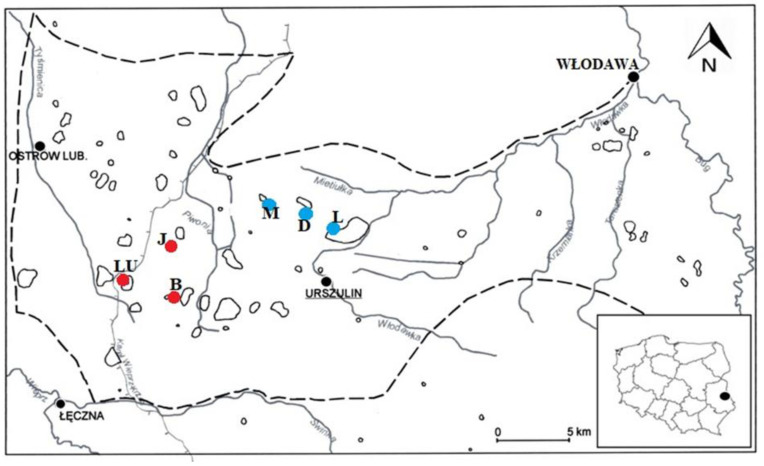
Location of water and plant material sampling stands: LU—Łukietek, J—Jelino, B—Brzeziczno, M—Moszne, D—Długie, L—Łukie. Blue dots—the location of natural aldrovanda sites, Red dots—sites where reintroduction of the species is planned, Black dots—location of villages and towns.

**Figure 3 ijerph-19-10743-f003:**
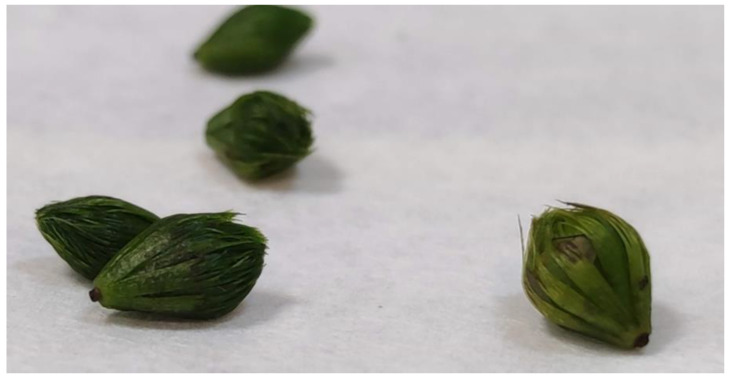
*Aldrovanda vesiculosa* turions taken from natural stands.

**Figure 4 ijerph-19-10743-f004:**
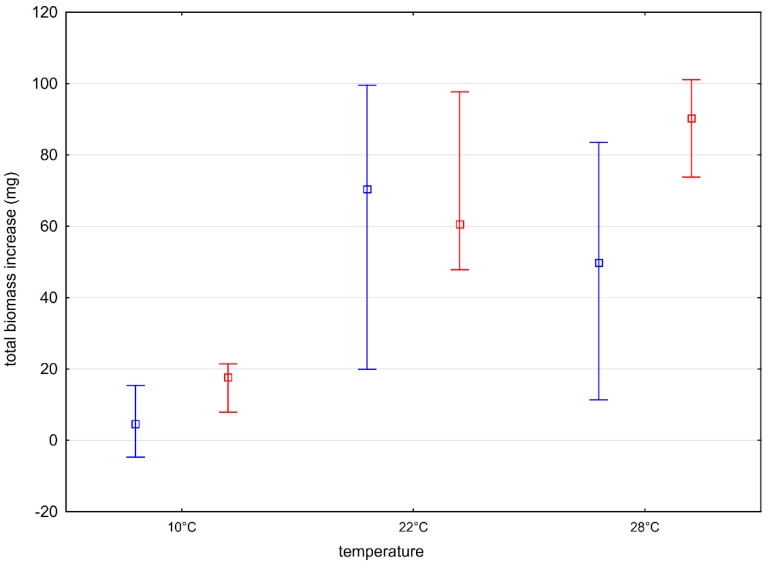
Increase in biomass (mg F.W.) of plants developing from turions under different thermal conditions. Plants from the natural site in Lake Długie are marked in blue, while plants from Lake Łukie are marked in red. Whiskers indicate the range of values, and squares indicate the median.

**Figure 5 ijerph-19-10743-f005:**
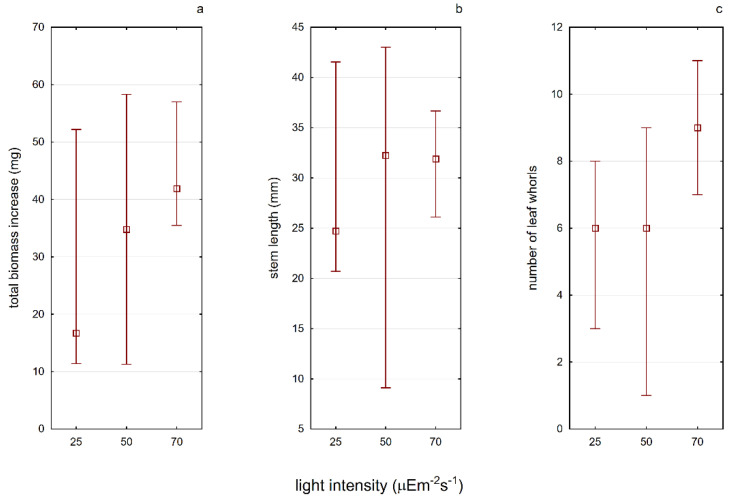
The influence of different light conditions on the tested plant characteristics ((**a**)—total biomass increase, (**b**)—stem length, (**c**)—number of leaf whorls) determining the efficiency of its growth in laboratory conditions. Whiskers indicate extreme values, and squares indicate the median.

**Figure 6 ijerph-19-10743-f006:**
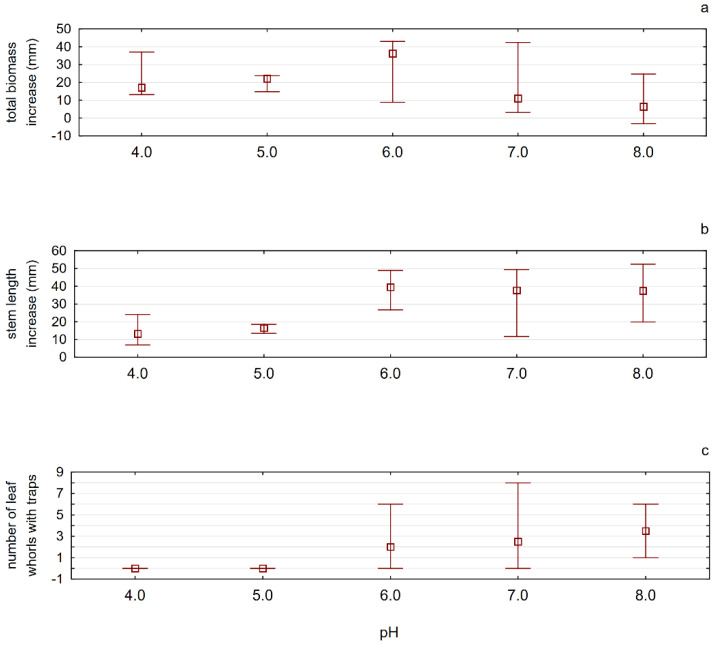
The influence of different values of the solution pH on the tested plant features, determining the effectiveness of its growth in laboratory conditions ((**a**)—total biomass increase, (**b**)—stem length, (**c**)—number of leaf whorls with traps). Whiskers indicate the range of values, and squares indicate the median.

**Figure 7 ijerph-19-10743-f007:**
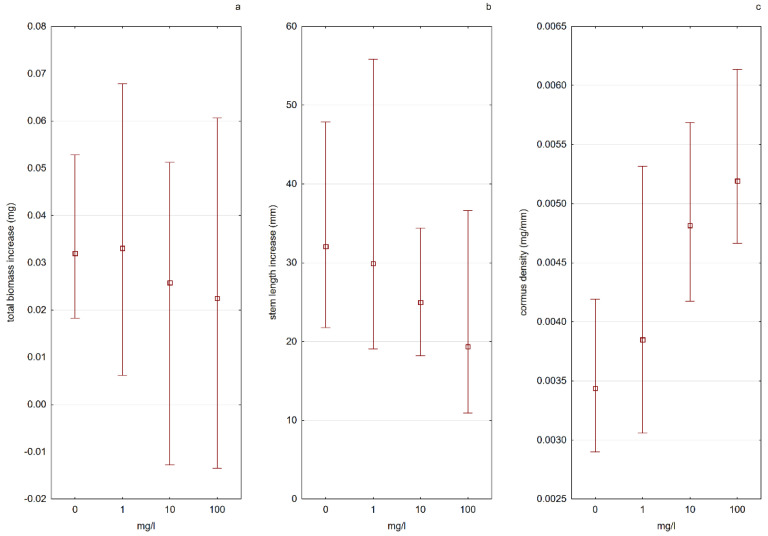
Influence of the content of available N-NO_3_ ions on the tested plant features, determining the growth efficiency ((**a**)—total biomass increase, (**b**)—stem length, (**c**)—cormus density). Whiskers indicate the range of values, and squares indicate the mean.

**Figure 8 ijerph-19-10743-f008:**
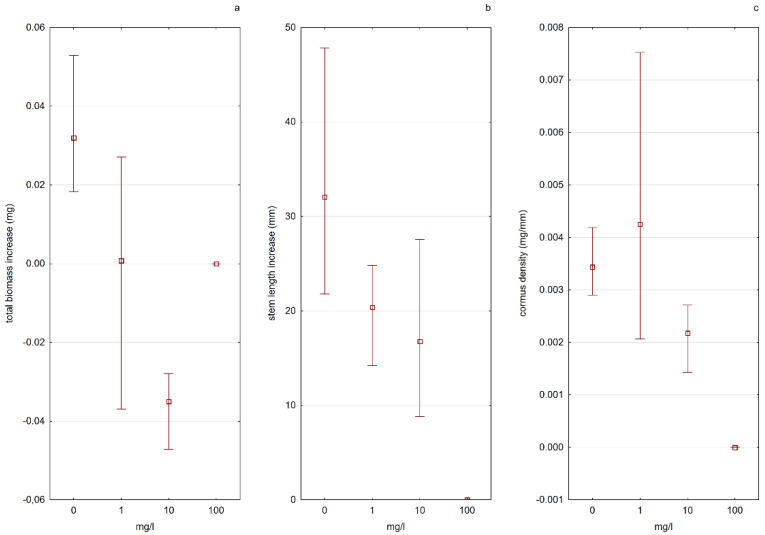
The influence of the available N-NH_4_ ions on the tested features of plants, determining the growth efficiency ((**a**)—total biomass increase, (**b**)—stem length, (**c**)—cormus density). Whiskers indicate the range of values, and squares indicate the mean.

**Figure 9 ijerph-19-10743-f009:**
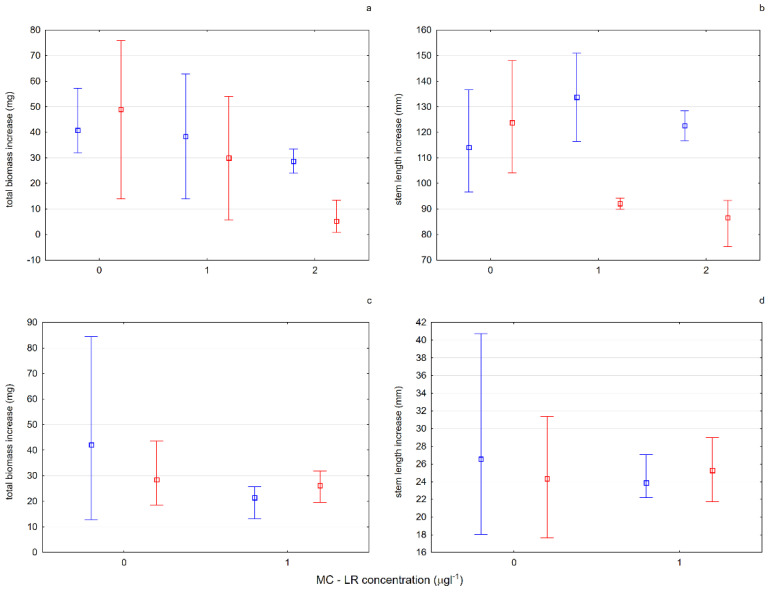
The effect of different concentrations of MC-LR on the biomass increase and the growth of the shoots from plants (**a**,**b**) or from turions (**c**,**d**). Plants from the natural site in Lake Łukie are marked in blue and those from Lake Długie in red. Whiskers indicate the range of values, and squares indicate the mean.

**Figure 10 ijerph-19-10743-f010:**
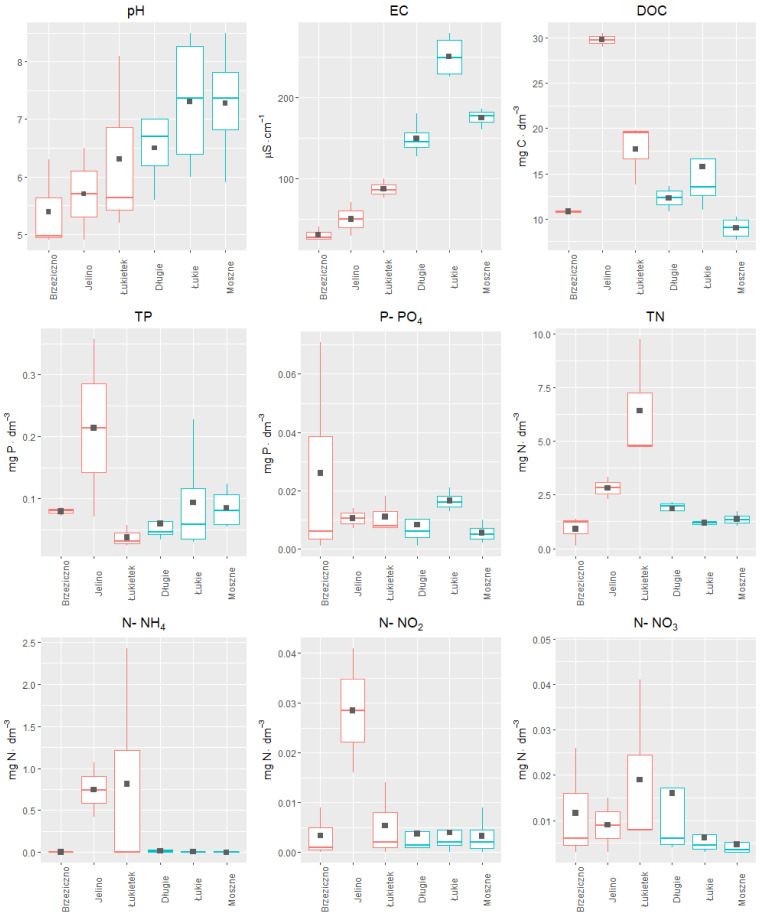
Distribution of values of the investigated physical–chemical factors of water at the study sites in 2019. The mean value of the data is marked by a filled square. The lakes with aldrovanda populations are marked in blue, and the reservoirs taken into account as substitute habitats for the species are marked in red.

**Figure 11 ijerph-19-10743-f011:**
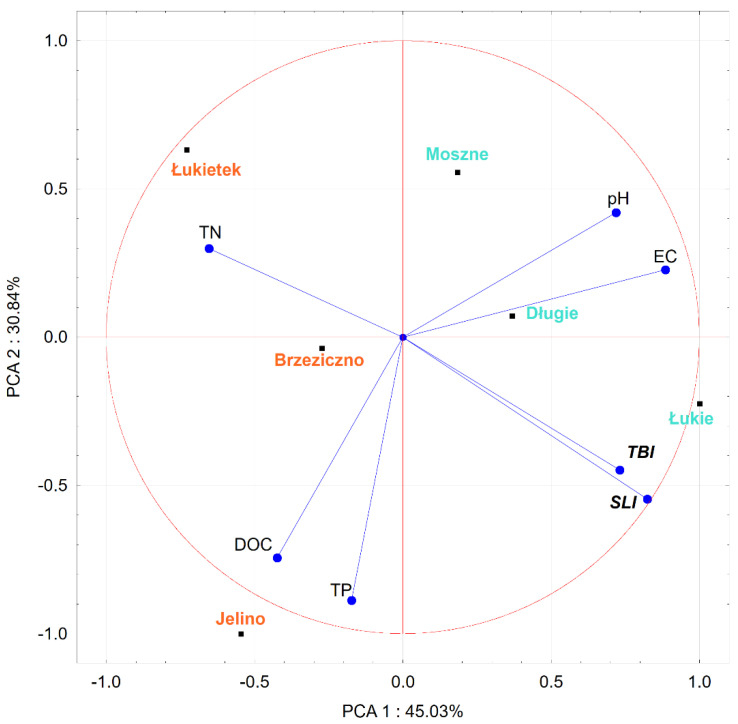
Principal component analysis biplot of the dataset. Reservoirs with aldrovanda populations are marked in blue, and those taken into account as substitute habitats for the species are marked in red. TBI—total biomass increase, SLI—stem length increase.

**Figure 12 ijerph-19-10743-f012:**
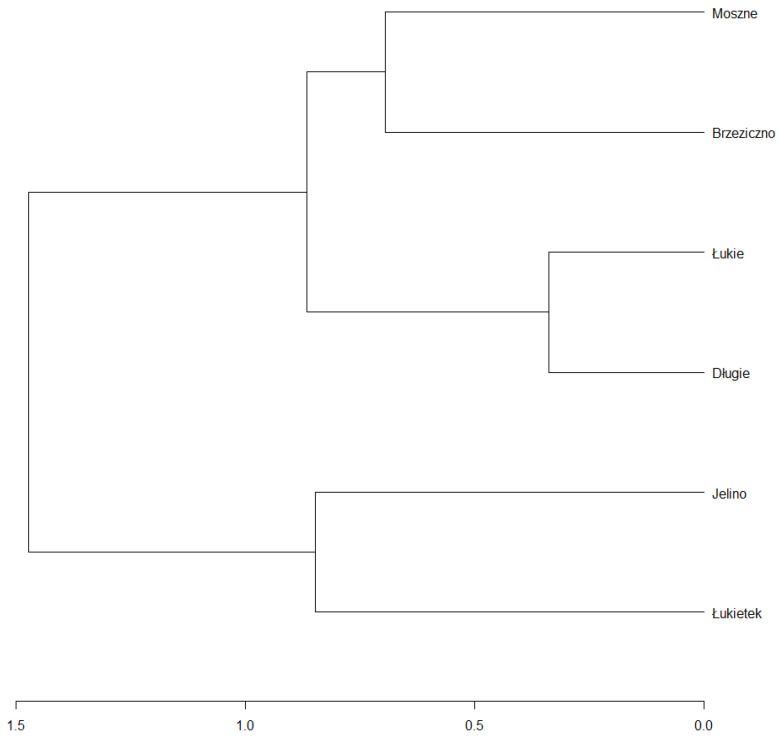
A dendrogram of the hierarchical cluster analysis based on binomial index and the average linkage method.

**Table 1 ijerph-19-10743-t001:** Characteristics of physical–chemical conditions of water from the studied lakes.

Sites	pH	EC	N-NH_4_	N-NO_3_	N-NO_2_	N_tot_	P-PO_4_	P_tot_	DOC
		µS·cm^−1^	mg·L^−1^						
Łukie (L)	6.52	225	0.001	0.003	0.0	1.228	0.017	0.037	13.1
Długie (D)	6.39	142	0.032	0.005	0.001	2.059	0.005	0.034	13.6

**Table 2 ijerph-19-10743-t002:** Content of photosynthetic pigments in *Aldrovanda vesiculosa* plant material in the chosen experiments (control—values at the beginning of the experiments; Chl*a*—chlorophyll *a*; Chl*b*—chlorophyll *b*, Car—carotenoids, ∑—sum of photosynthetic pigments; Chl*a/b*—ratio of chlorophyll *a* to *b*).

Experiments	Variants	Chl*a*		Chl*b*		Car (μg/g)		∑	Chl*a/b* Ratio
		Mean	SD	Mean	SD	Mean	SD		
Control	0	518.23	113.19	303.50	120.93	421.66	28.09	1243.39	1.708
E2	25 µEm^−2^s^−1^	596.44	76.41	306.25	80.53	286.76	39.38	1189.45	1.948
	50 µEm^−2^s^−1^	626.64	66.65	432.63	98.31	283.99	25.19	1343.26	1.448
	70 µEm^−2^s^−1^	705.01	105.13	353.83	71.59	350.79	77.59	1409.63	1.993
E3	pH 4.0	831.48	31.33	383.99	53.97	377.02	27.00	1592.49	2.165
	pH 5.0	959.40	158.34	496.20	143.24	386.07	21.06	1841.67	1.933
	pH 6.0	646.65	268.83	359.67	222.92	319.81	29.82	1326.13	1.798
	pH 7.0	951.61	213.42	511.99	226.09	464.97	78.47	1928.57	1.859
	pH 8.0	1018.14	157.87	471.05	105.17	476.95	133.13	1966.14	2.161
E5 turions	L 0	737.16	213.01	439.70	198.39	290.39	30.28	1467.25	1.677
	L 1	616.41	77.51	285.97	31.61	340.39	51.46	1242.77	2.156
	D 0	711.82	123.29	393.23	47.78	355.19	35.66	1460.24	1.810
	D 1	694.67	77.08	327.06	61.40	298.01	4.30	1319.74	2.124
E5 plants	L 0	462.38	93.43	230.30	39.16	216.28	60.21	908.96	2.008
	L 1	420.54	62.20	197.77	32.54	219.53	29.20	837.84	2.126
	L 2	338.43	73.99	149.42	23.68	167.28	25.24	655.13	2.265
	D 0	492.32	57.53	253.00	30.93	229.23	18.78	974.55	1.946
	D 1	531.91	105.38	247.13	10.27	255.53	66.71	1034.57	2.152
	D 2	518.60	100.75	271.29	85.79	257.45	22.92	1047.34	1.912

## Data Availability

All the required data related to the current study are embedded in this manuscript. If needed, the data are available upon request from the corresponding author.

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
