# Peer review of "Searching for Optimal Substitute Habitats for Plants by Biological Experiments—A Case Study of the Endangered Species Aldrovanda vesiculosa L. (Droseraceae)"

_ijerph, 2022, doi:10.3390/ijerph191710743_

Round 1

Reviewer 1 Report

The quality of the manuscript is good, and the methods employed are coherent. However authors need to improve the discussion trying to present better the results and the importance of the study.

Reviewer 2 Report

Authors of the manuscript “Searching for optimal substitute plant habitats by biological experiments - a case study of the endangered plant species Al- drovanda vesiculosa L. (Droseraceae)” revealed that that both physical chemical parameters of water to  gether with requirements of the species, are crucial for the choice of the appropriate sites for plants translocation. However, the observations made by the practitioners are also important, as they are able to recognize the problems which are not measurable by typical methods.

Title of this study suits the journal; results are presented and discussed in a nice way. However authors have to address following minor issues before the possible publication of this article:

1.      Please add important findings of current study in the abstract section.

2.      Please combine short paragraphs at the appropriate places with other paragraphs.

3.      L94-95: Please rewrite the sentence.

4.      Please delete unnecessary detail from the introduction section.

Reviewer 3 Report

Comments: Minor revision

Abstract needs revision as some sentences are vague like ‘The presented manuscript describes the experiments conducted to make such a selection of sites on the 18 example of Aldrovanda vesiculosa, a plant species critically endangered with extinction’.

There is a repetition in keyword section

In introduction section the sentence, ‘The species is facing a potential threat in its natural 56 habitat due to changing environmental factors such as eutrophication and climate alterations’ needs Justification with suitable citations.

Some sentences needs rephrasing.

Material and Methods needs incorporation of some relevant citations

Better to write the results and discussion sections separately, if possible

Conclusion must include the key finding of the study along with the its significance
